# Yangtze River Basin Environmental Regulation Efficiency Based on the Empirical Analysis of 97 Cities from 2005 to 2016

**DOI:** 10.3390/ijerph18115697

**Published:** 2021-05-26

**Authors:** Qian Zhang, Decai Tang, Brandon J. Bethel

**Affiliations:** 1School of Economics and Management, Nanjing Forestry University, Nanjing 210037, China; 2School of Law and Business, Sanjiang University, Nanjing 210012, China; 3School of Marine Sciences, Nanjing University of Information Science & Technology, Nanjing 210044, China; 20195109101@nuist.edu.cn

**Keywords:** Yangtze River Basin, environmental regulation efficiency, SE-SBM model, DEA–Malmquist index

## Abstract

The Yangtze River Basin (YRB) is an important area for China’s economic development and environmental governance. The aim of this paper is to analyze the total factor productivity across 97 cities in the YRB from 2005 to 2016. Based on the input and output indicators from 2005 to 2016, this paper selects the SE-SBM model to measure the environmental regulation efficiency (ERE) of 97 cities in the YRB and then uses the DEA–Malmquist index to measure the total factor productivity of the region. Results suggest that the overall ERE in the YRB is weakly ineffective, while ERE in the central and eastern coastal areas is relatively high. ERE matches the economic foundation and development of the city. YRB environmental regulation efficiency was in descending order in the middle stream, upstream, and downstream. The efficiency of regional environmental regulation shows an N-type development trend, with obvious characteristics of phased development. Moreover, the total factor productivity of the YRB has shown a downward trend. The scale efficiency index and the technical efficiency index have positively boosted the total factor productivity, while the technological progress index has dragged down the total factor productivity of the area. The contribution to the total factor productivity index is in order of scale efficiency, technological progress index, and technological efficiency index in the downstream. The overall inputs and outputs of the YRB have great development potential. The inputs have not been fully utilized, the outputs have not been maximized, and the regional differentiation is significantly observable.

## 1. Introduction

The Yangtze River Basin (YRB) contains a vast array of developed industries in densely populated cities and rich mineral, water, and agricultural resources; nearly half of the country’s heavy chemical, power, and steel companies are distributed along the river, and as such, environmental protection is an important facet of further development. As a result, the “Master Plan for the Protection, Development and Utilization of the Yangtze River Coastline” issued in September 2016 comprehensively analyzed the main problems in the development and utilization of both sides of the Yangtze River. China’s 14th Five-Year Plan in 2020 emphasizes the environmental issues and will strive to coordinate and integrate resources and promote the green development of both sides of the Yangtze River. Considering these realities, the aim of this paper is to study total factor productivity across 97 cities in the YRB from 2005 to 2016.

In recent years, China has issued a series of environmental protection policy documents, and many scholars have begun to pay attention to the efficiency of environmental regulations. Yu and Wang [1] argued that China is committed to strengthening the construction of environmental laws and regulations to optimize the structure of the industrial economy and achieve high-quality economic development. Recent research has shown that the overall level of China’s regional environmental efficiency is relatively low, with large regional differences but gradually showing signs of improvement [2]. Environmental regulation efficiency (ERE) in China shows a clear “polarized effect”, matches the level of economic development [3], and is often studied using data envelopment analysis (DEA) models. A three-stage DEA model and the super DEA–Malmquist method were widely used to measure pollution treatment efficiency in China [4]. Wang et al. [5] used network DEA to study the ERE of China’s five major urban agglomerations from 2000 to 2014 and found wave-like growth where low environmental regulatory efficiency was a primary reason for increasing industrial efficiency. Amongst the studied agglomerations, the Yangtze River Delta (YRD) and the Beijing–Tianjin–Hebei regions had the highest overall industrial efficiency. The authors also noted that if the ecological environment is to be improved, improving the efficiency and technical level of industrial pollution control is crucial. Based on the two-stage, network-based super-efficient data envelopment analysis (DEA) approach, the efficiency of China’s industrial environmental regulation improved from 2004 to 2015, though room for further improvement exists [6]. Tang and Bethel [7] used the super efficiency DEA and Malmquist index to study environmental remediation in the YREB from the perspective of input–output optimization from 2003 to 2013 and found that efficiency was not only low but was deteriorating. These results confirm earlier observations of Chen and Jia [8] that also found a downward trend of environmental efficiency. That study also identified that the level of management and scale optimization was the main factor that inhibited total factor productivity growth. Industrial governance efficiency is better than production efficiency; heavy industry and high energy-consuming industries are the main reasons for the overall low efficiency. The ERE of China’s steel industry has been very low in the past ten years, and the efficiency of all-factor environmental governance has shown a downward trend from 2005 to 2014, mainly due to the technological progress change index [9]. High-tech industries not only have high overall efficiency but can also achieve a win–win situation between their performance and national contribution. The agglomeration of high-tech industries can improve the regional ecological environment, and it can more effectively improve the environmental regulation efficiency in economically developed areas [10]. Pan et al. [11] in an earlier study identified that there is no significant difference between improving efficiency through environmental regulations and market incentives in the short term, and market incentives can more effectively improve environmental regulation efficiency in the long run. The abundance of resources can also affect the level of ERE. The economically developed areas are conducive to ERE improvement due to resource agglomeration, while the economically backward areas are likely to be ineffective due to lack of resources. The efficiency of central cities is relatively high, and the ecological efficiency of urban agglomerations has significant differences [12]. Peng et al. [13] identified that from 2012 to 2016, the overall level of environmental governance of the YRD greatly improved, but from a dynamic perspective, pollution generated during urbanization degrades urban environmental governance performance. Song and Wang [14] and Peng et al. [15] found that due to technological progress, significant gaps between the efficiencies of urban governance in developing countries has declined sharply in the past ten years. China’s energy utilization and environmental efficiency are low, pollution emissions need to be greatly improved, and environmental supervision costs are high. Advanced technologies should be used to improve ERE [16]. In terms of the selection of input and output factors, inputs are mainly divided into two categories, one is pollution control input, the other is total labor and capital inputs; output is mainly good output and bad output. The good output is mainly economic growth and ecological improvement. Bad output is usually measured by wastewater, exhaust gas, and solid waste. The innovation of this article is in the selection of output variables; in the context of global climate governance, forestry and temperature are included. In the method, the SE-SBM method is adopted, which not only analyzes the super-efficiency value, but also analyzes the efficiency of each input and output in detail in conjunction with the slack value. In terms of research objects, we did not stay at the provincial level but went to the prefecture–city level. Data of 120 cities were collected. Due to the completeness of the data analysis, 23 cities were proposed, and data of 97 cities were finally left. The rest of the paper is structured as follows. Section 2 and Section 3 describe the methodology and dataset employed. Section 4 presents the main results, and Section 5 summarizes the main findings of this study and concludes.

## 2. Methodology

### 2.1. Study Area

The Yangtze River Basin (YRB) refers to the vast area through which the mainstream and tributaries of the Yangtze River flows through three economic areas of eastern, central, and western China. The mainstream of the Yangtze River flows through 11 provinces, autonomous regions, and municipalities that are directly controlled by the Central Government and includes Qinghai, Tibet, Yunnan, Sichuan, Chongqing, Hubei, Hunan, Jiangxi, Anhui, Jiangsu, and Shanghai. In addition to the provinces mentioned above, there are the Guizhou, Zhejiang, Shaanxi, Gansu, and Guangxi provinces into which the Yangtze River’s first-level tributaries flow. The YRB also includes Henan, Fujian, and Guangdong, which the mainstream of the Yangtze River and the first-level tributaries do not flow through, but some areas belong to the YRB, which has a total of 17 provinces, autonomous regions, and two municipalities directly under the Central Government. The basin covers an area of 1.8 million km^2^ and accounts for approximately 20% of China’s total land area. The Yangtze River is a hub of economic development, accounting for almost 50% of the country’s total economic output, covering three major economic regional urban agglomerations in China, including the Chengdu–Chongqing, Yangtze River middle reaches, and the Yangtze River Delta (YRD) urban agglomerations. Considering the availability of data, this paper finally selected 97 cities as the research objects (Figure 1).

### 2.2. The SE-SBM Model

The data envelopment analysis (DEA) method is a non-parametric technical efficiency analysis method based on the comparison between the evaluated objects. Using DEA for efficiency evaluation can obtain a lot of management information with economic connotation and background [17]. DEA measures and evaluates the input and output efficiency of each decision-making unit (DMU) through functional calculation tools and compares the efficiency of a specific unit with the efficiency of similar units that provide the same service. To solve the possible slack problem of input and output and the problem that multiple DMU efficiency values are the same, this paper uses the super-efficiency non-oriented slacks-based model (SE-SBM) with the variable returns to scale (VRS) assumptions to measure the efficiency of environmental regulation [18]. Compared with the traditional SBM model, the environmental regulation efficiency (ERE) calculated by the SE-SBM model is not limited to 1, which can effectively improve the comparability of the calculation results; at the same time, it can avoid the radial and angular differences. The deviation and influence of the results can better reflect the essence of efficiency evaluation [19]. If the ERE is greater than or equal to 1, it is effective. The larger the score, the higher the efficiency. If the ERE is greater than or equal to 0.5 and less than 1, then it is weakly inefficient, and if the ERE is less than 0.5, then it is strongly inefficient [3]. In the direction of ERE measurement, there are usually input-oriented models and output-oriented models. This article is more about the output efficiency, so we chose the latter to facilitate the calculation of TFP [20,21]. The output-oriented SE-SBM method can be formulated as:(1)$$min\rho_{se}=1/\left(1-\frac{1}{q}\sum_{r=1}^{q}S_{r}^{+}/O_{rk}\right)\;$$
(2)$$s.t.\sum_{j=1,j\neq k}^{n}I_{ij}\lambda_{j}\leq I_{ik}$$
(3)$$\sum_{j=1,j\neq k}^{n}O_{rj}\lambda_{j}+S_{r}^{+}\leq O_{rk}$$
where *n* is the number of decision-making units in the input–output system, *I*,*O* represent the vectors of input and output, respectively, m, q represent the number of variables, and λ is the weight vector. *j* = 1,2 … *n*, *i* = 1,2 … m, *r* = 1,2 … q. The K decision-making unit, ρ, is the efficiency value. $\overset{n}{\underset{j=1}{\sum }}\lambda_{j}O_{ij}\;$ and $\overset{n}{\underset{j=1}{\sum }}\lambda_{j}O_{rj}$ are virtual DMUs, namely benchmark data, (*I_k_*, *O_k_*) are the evaluated DMUs, and S represents the slack value. The DEA model is composed of multi-segment linear functions. If a single DMU projection falls within the parallel section, the problem of slack variables will occur. The measurement in the radial model does not consider the slack variables. The advantage of the SE-SBM non-radial model is that it can judge the impact of each input and output on the overall efficiency based on the slack value. In general, the slack value can be obtained by adding slack variables to the constraints of the model [18]. If the values of the slack variable are 0, it means that the input has been fully utilized and the output has been maximized. If the slack value is not 0, this indicates that there is still great potential for development in input and output. The excessive proportion of redundant values in the slack variable indicates that the input or output efficiency of this element is low. 

### 2.3. DEA–Malmquist Index

Originally proposed by Malmquist in 1953, this Malmquist index (MI) was originally suitable for measuring changes in production efficiency. Färe et al. [22] used the index in combination with the DEA method of non-parametric models to better measure the production efficiency of multi-input–output and panel data. The Malmquist index can be decomposed into the technical progress index and technical efficiency change index. If scale efficiency is variable, then the technical efficiency change index can be further decomposed into pure technical and scale efficiencies [23]. The traditional DEA model is to measure the static relative efficiency of different decision-making units in the same period, that is, the overall technical efficiency change, while the Malmquist index model is an analysis of the dynamic efficiency of the data of each decision-making unit in different periods. We use DEA-Solver_pro13.1 software (SAITECH, Tokyo, Japan) to calculate the MI, which is the geometric mean instead of arithmetic averages. The formula for the MI index is as follows:(4)$$M\left(I^{t+1},O^{t+1},I^{t},O^{t}\right)=\frac{D^{t}\left(I^{t+1},O^{t+1}\right)}{D^{t}\left(I^{t},O^{t}\right)}\times \sqrt{\left[\frac{D^{t}\left(I^{t+1},O^{t+1}\right)}{D^{t+1}\left(I^{t+1},O^{t+1}\right)}\times \frac{D^{t}\left(I^{t},O^{t}\right)}{D^{t+1}\left(I^{t},O^{t}\right)}\right]}$$
where *I* and *O* represent input and output, respectively, *t* represents time, and *D* represents distance function. If MI is greater than 1, the productivity will show an upward trend from *t* to *t* + 1. Conversely, MI less than 1 indicates that productivity tends to decrease. If it is equal to 1, it means the productivity remains unchanged from *t* to *t* + 1 [24].

The environmental TFP index (MI) can be decomposed into the technical efficiency index (EFF_ch_) and the technological progress index (TP_ch_). If the scale efficiency is variable, the technical efficiency index (EFF_ch_) can be further decomposed into pure technical efficiency (PE_ch_) and scale efficiency (SE_ch_). The relationship is as follows:TFP_ch_ = EFF_ch_ × TP_ch_ = Catch-up × Frontier-shift (Innovation)(5)
EFF_ch_ = PE_ch_ × SE_ch_(6)

EFFch is calculated based on the output-oriented CCR model and includes the effect of scale efficiency. Its essence is maxθ, which means maximize the efficiency value, and this function is subject to the following relationship:$$\sum_{j=1}^{n}\lambda_{j}O_{ij}+S_{i}^{-}=I_{ij};\;\sum_{j=1}^{n}\lambda_{j}O_{rj}+S_{r}^{+}=\theta O_{rk};\;\lambda \geq 0,\;S^{-}\geq 0,\;S^{+}\geq 0$$

PEch is calculated based on the VRS output-oriented BCC model. Because the influence of scale is excluded, it is also called “pure technical efficiency”. Unlike CCR, its benchmark is at the forefront of pure technical efficiency. BCC added constraints $\overset{n}{\underset{j=1}{\sum }}\lambda_{j}=1$ on the basis of CCR. Its essence is the ratio of the productivity of the evaluated DMU to the productivity of the reference benchmark. If the value is greater than 1, it will have a positive effect on TFP, if it is less than 1, it may drag down the TFP. EFF_ch_ mainly reflects whether production input elements are effectively used or whether resources are reasonably allocated. TP_ch_ mainly reflects innovation and technological progress. PE_ch_ is mainly affected by system differences and management levels. SE_ch_ is mainly influenced by the structure and scale of resource allocation.

## 3. Statistical Datasets

Considering the applicability and availability of data, this paper finally selected 97 prefecture-level cities in the Yangtze River Basin from 2005 to 2016, where all input and output data samples were acquired from the China Regional, Economic, Forestry, Industrial Economic statistical yearbooks. Additional data were derived from climate bulletins of each city and province. Interpolation was used to eliminate gaps where an index with a value of 0 is artificially assigned a very small positive value to facilitate model calculation. Drawing on the selection of indicators in the existing literature [25,26], this paper selects labor input, physical capital input, and human capital input. The outputs conclude economic growth, ecological environment, pollution, and climate (Table 1) to promote transformational development and improve the ecological environment.

## 4. Results

### 4.1. Yangtze River Basin Environmental Regulation Efficiency

The environmental regulation efficiency (ERE) in YRB during 2005–2016 is 0.6239, which is weak and invalid. The areas with higher environmental regulation efficiency are mainly concentrated in the central and eastern coastal areas of the basin (see Figure 2a for details). From the perspective of cities, 23 of the 97 cities have an environmental regulation efficiency greater than 1, which are Changde, Zhangjiajie, Xinyu, Yingtan, Ganzhou, Suzhou, Shanghai, Quzhou, Xuancheng, etc. The highest rate in Chongqing is 1.44, indicating that environmental regulations are very effective. There are 31 cities with ERE between 0.5 and 1, including Hengyang, Shaoyang, Yueyang, Yiyang, Chenzhou, Yongzhou, Huaihua, Nanchang, Pingxiang, Jiujiang, etc., indicating that the environmental regulation efficiency in these cities is weakly ineffective. There are 43 cities with ERE lower than 0.5, which includes Changsha, Zhuzhou, Xiangtan, Jingdezhen, Changzhou, Nantong, Yangzhou, Taizhou, Jiaxing, Huzhou, Wuhu, Huzhou, and other cities. The environmental regulations of these cities are strongly ineffective (see Appendix A for details). Compared to 2005, ERE in 49 cities has improved, and ERE in the remaining 48 cities has declined. The main reason lies in the high efficiency of environmental regulations in yellow areas (Figure 2b). However, cities with lower environmental regulation efficiency have more room for improvement, which is consistent with the theory of economic convergence [27].

From 2005 to 2016, we can see that Chongqing, Yingtan, Zhangjiajie, and Bazhong rank the top four among the cities in terms of environmental regulation efficiency, all of the four cities are located in the middle and upper reaches of YRB (Figure 3). The average efficiency of urban environmental regulation in the upstream of YRB is 0.58, the middle stream is 0.77, and the downstream is 0.52.

Figure 4 shows that the middle steam cities have the highest environmental regulation efficiency, followed by the upstream cities, and the efficiency of the downstream cities is the lowest. These three regions present an N-type waveband development trend. From 2005 to 2006, the three regions all showed an upward trend, and the period 2007 to 2012 showed a downward trend. Subsequently, the efficiency of environmental regulations showed a clear upward trend. This is mainly due to the relevant environmental governance policies issued by the state. However, the overall ERE is basically in a state of inefficiency or even ineffectiveness.

### 4.2. DEA–Malmquist Index

From 2005 to 2016, the overall productivity of YRB showed a downward trend based on geometric average, and TFP was 0.93. However, the arithmetic average result was 1.06, which showed an upward trend. The arithmetic average method made the Malmquist index result biased (Section A.1). Among these cities, 28 cities have shown increasing productivity, such as Changsha, Zhuzhou, Chenzhou, Xinyu, Shangrao, etc., which are mainly located in the lower reaches of YRB and part of the middle reaches. The total factor productivity of the remaining cities showed a downward trend in varying degrees during this period (Figure 5a). During this period, 50 cities have achieved an increase in the total factor growth rate, while the remaining cities have experienced a decline in productivity. Figure 5b shows the change in the total factor growth rate of each city in 2005 and 2016. A value greater than 0 indicates that productivity has improved, the darker the color, the greater the progress. Values less than 0 mean that the productivity has fallen back compared with 2005.

From 2005 to 2016, the productivity index of YRB showed a wave-like trend. From 2005 to 2007, 2009 to 2010, and 2013 to 2014, the productivity value increased. Displayed in Figure 6, it can be observed that the productivity of the downstream is generally higher than that of the upstream and midstream, and the fluctuation of the midstream is the smallest. From 2007 to 2009, 2011 to 2013, and from 2015 to 2016, however, the value of productivity declined. This is related to the relaxation of environmental protection after the 2008 Olympic Games [28]. The Olympic Games had a positive impact on the environmental efficiency of the Beijing area but had a negative impact on the surrounding areas. The hosting of the Beijing Olympics has greatly increased investment in infrastructure, but it has also increased carbon dioxide emissions and reduced environmental efficiency [28]. From 2001 to 2007, the overall utilization rate of China’s industrial capacity continued to rise but tended to decline from 2008. After 2008, the utilization rate of the steel industry dropped from 80 to 71% in 2014 and 66.99% in 2015. In 2014, steel production capacity declined to 1.14 billion tons in 2016. China’s economy officially entered a new stage of consumption upgrading. Life has shifted from a subsistence life to a well-off level. Data from the China Labor Statistics Yearbook show that the average working hours of urban employees dropped from 46.6 to 46.1 h per week from 2014 to 2016, and the proportion of employed persons in the total population dropped from 56.5 to 56.1%. The decline in labor input and industrial production capacity ultimately led to a decline in TFP from 2015 to 2016. Besides, Hunan is a key province of non-ferrous metals [29]. In 2014, heavy metals in the Xiangjiang River Basin (part of YRB) where Hunan is located have seriously exceeded the standard, and industrial pollution discharge has become an important source of environmental pollution [30].

### 4.3. TFP Decomposition

The overall TFP value of YRB is 0.93, which is specifically decomposed into a EFF_ch_ of 1.00, a PE_ch_ of 0.93. The technical progress index of further decomposed into PE_ch_ of 0.94 and Se_ch_ of 1.07. The input of production factors in the entire region can basically be effectively used, and the resource allocation structure is relatively reasonable, but there are still insufficient innovations, serious regional differentiation, and uneven management levels. YRB lacks specificity in response to local conditions. Meanwhile, the YRB has a large gap in institutional guarantees, technological innovation, and management control and has not yet formed an efficient management pattern (see Section A.2 for more details).

The environmental TFP indices of the upper and middle reaches of the Yangtze River were 0.89 and 0.90, respectively, which means that total factor productivity did not rise from 2005 to 2016 but showed a downward trend. The EFF_ch_ of upstream and middle stream has shown an upward trend, both approximately unity. SE_ch_ has also been improved, but the PE_ch_ and SE_ch_ are both less than 1, indicating the technological innovation, institutional system, and management level of both areas of YRB need to be improved. The downstream, which mainly includes the YRD, has achieved the factor productivity of 1.07, showing a clear upward trend. Decomposing the TFP, EFF_ch_, TP_ch_, and SE_ch_ are 1.01, 1.06, and 1.08, respectively. It shows that the overall technical efficiency of the lower reaches of the Yangtze River is relatively high, the resource allocation is relatively reasonable, and the city’s ability to coordinate development is relatively strong. Among them, the scale efficiency index contributes the most to total factor productivity. However, as shown in Figure 7, pure technical efficiency is less than 1, indicating that there are certain disparities in urban development, and there is much room for improvement of environmental regulation management.

### 4.4. Slack Analysis Based on SE-SBM Model

Looking at the Yangtze River Basin as a whole, from 2005 to 2016, the efficiency of inputs and outputs was not high, and regional differentiation was large. There is much room for improvement in the input and output efficiency of environmental regulations in YRB. In terms of inputs, 19 cities made full use of labor input, including Pingxiang, Ganzhou, Wuxi, Changzhou, Suzhou, Zhenjiang, Shanghai, Nanping, Longyan, Maanshan, Tongling, Huangshan, Deyang, Ziyang, Nanyang, Zhumadian, Chongqing, etc. There are 25 cities in which capital is fully utilized, including Yueyang, Xinyu, Suzhou, Shanghai, Hangzhou, Quzhou, Chongqing, etc. There are 14 cities with the most efficient human capital input, including Changde, Xinyu, Ganzhou, Wuxi, Suzhou, Shanghai, Quzhou, etc. Overall, the seven cities where the inputs are most effectively used are Suzhou, Shanghai, Nanping, Longyan, Nanyang, Zhumadian, and Chongqing. In terms of outputs, 27 cities have the highest economic output efficiency, mainly including Zhuzhou, Xiangtan, Yueyang, Zhangjiajie, Nanjing, Nantong, Yangzhou, Zhenjiang, Huzhou, Nanping, Wuhu, Wuhan, etc. In terms of ecological indicators, there are 11 cities with high efficiency in artificial afforestation area and forestry output value, including Changde, Zhangjiajie, Xinyu, Yingtan, Suzhou, Shanghai, Quzhou, Longyan, Suizhou, Guangyuan, and Zhumadian. There are 14 cities with high efficiency in the treatment of industrial wastewater discharge, including Zhangjiajie, Shanghai, Fuzhou, Suizhou, etc., and 23 cities with higher efficiency in the treatment of agricultural land pollution, including Shaoyang, Zhangjiajie, Yiyang, Suzhou, Nantong, Taizhou, and other cities. In terms of climate governance, the overall efficiency is low, except for Enshi and Zhumadian (Figure 8).

From the perspective of time development, in 2005, the labor input of 49 cities was maximized, the material capital input of 64 cities was effective, and the education input of 45 cities was fully utilized. In 2016, these three indicators were 60, 53, and 44, respectively. In 2005, there were 27 cities with effective overall input of factors. In 2016, the number rose to 31. In 2005, 65 cities maximized economic output and 63 in 2016, a slight decrease. In terms of ecological construction, 48 cities had the most efficient output of planted forests and 51 cities in 2016. In 2005, the forestry output value was the most effective, and there were 61 cities, but 63 cities in 2016. In 2005, there were 48 cities with the most effective industrial wastewater treatment and 52 cities in 2016; in terms of agricultural land pollution treatment, there were 58 cities in 2005 and 66 cities in 2016. As for climate governance, 42 cities were the most effective in 2005 and 38 cities in 2016. The difference gap between cities is obvious.

## 5. Conclusions and Recommendations

The overall efficiency of environmental regulations in the YRB is weakly ineffective, while the efficiency of environmental regulations in the central and eastern coastal areas is relatively high. From 2005 to 2016, half of the urban areas improved environmental regulation efficiency, while the other half showed a downward trend. The efficiency of environmental regulation matches the initial economic conditions and development. Over the period of 2005 to 2016, environmental regulation efficiency in the middle stream of the Yangtze River, the upstream of the Yangtze River, and the downstream of the Yangtze River were ranked in descending order. The regional environmental regulations efficiency shows an N-shaped trend with obvious characteristics of phased development. The first high point appeared in 2007, and the second low point appeared in March 2012. The total factor productivity of YRB showed an overall downward trend from 2005 to 2016, but 28 cities showed an upward trend, and these cities were mainly located in the middle and downstream of the Yangtze River. During this period, the productivity showed a double M-shaped trend, among which the TFP value peaked at 2007, 2011, and 2015, respectively. The scale efficiency index and the technical efficiency index have positively promoted total factor productivity, while the technological progress index has dragged down the total factor productivity of the Yangtze River Basin. The contribution to the downstream total factor productivity index is in order of scale efficiency, technological progress index, and technological efficiency index. From 2005 to 2016, the overall input and output efficiency of the YRB had great development potential. The inputs have not been fully utilized, and the outputs have not been maximized. The input–output efficiency varies greatly among cities. In 2016, 31 cities’ inputs were fully utilized; only 18 cities maximized their output, and 17 cities achieved better results in pollution control and climate control. Compared with 2005, the number of cities that fully utilized labor input factors increased in 2016, but the cities that fully utilized education factors remained almost unchanged, while the number of cities that fully utilized physical capital decreased. In 2016, the number of cities that maximized economic output decreased slightly. Ecological construction and pollution control were effective, but climate control in the whole area was still ineffective.

Based on the above conclusions, the following suggestions can be made. Firstly, investments in material and human capital for pollution control should be increased to improve high-quality economic development and promote green development. The expected outputs can be enhanced, and undesired output must be well controlled. The problem of “chemical encirclement of the river” should be solved as soon as possible to reduce the total amount and intensity of pollution. Secondly, environmental regulation policies should be tailored to local conditions. Based on the differences in environmental resource carrying capacity and economic conditions among these cities, the government should establish a top-level design and management mechanism for environmental policies. The ability of regional coordinated development and management mechanisms’ systemic control needs to be strengthened. Thirdly, enterprises should insist on scientific and technological innovation and make full use of big data, geographic information system, and high-tech environmental protection technology in pollution control and environmental management and also establish an early warning mechanism to improve the refined management efficiency.

## Figures and Tables

**Figure 1 ijerph-18-05697-f001:**
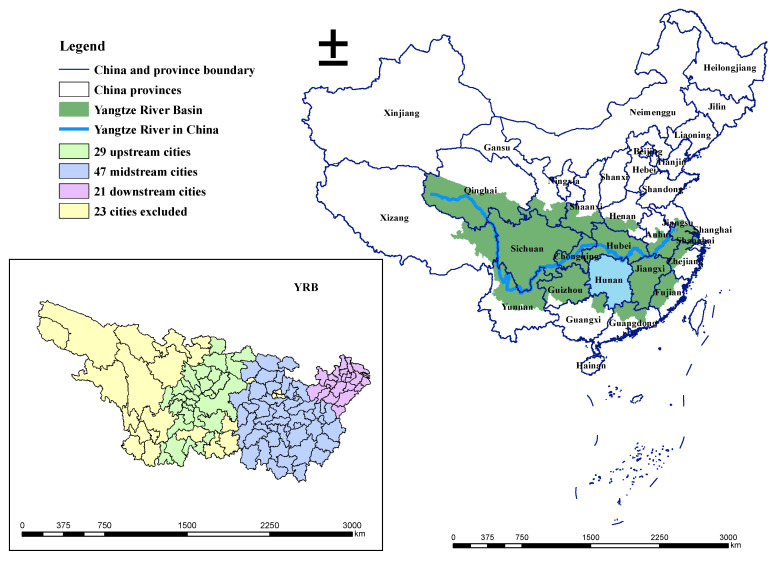
The Yangtze River Basin study area.

**Figure 2 ijerph-18-05697-f002:**
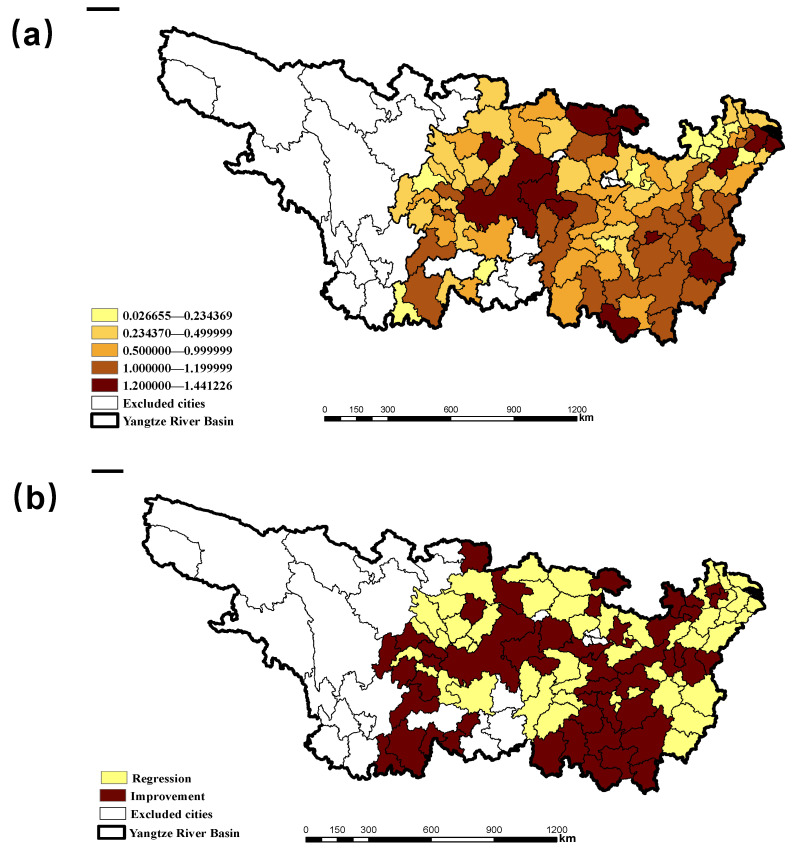
ERE (**a**) and its change (**b**) of 97 cities in YRB during 2005–2016.

**Figure 3 ijerph-18-05697-f003:**
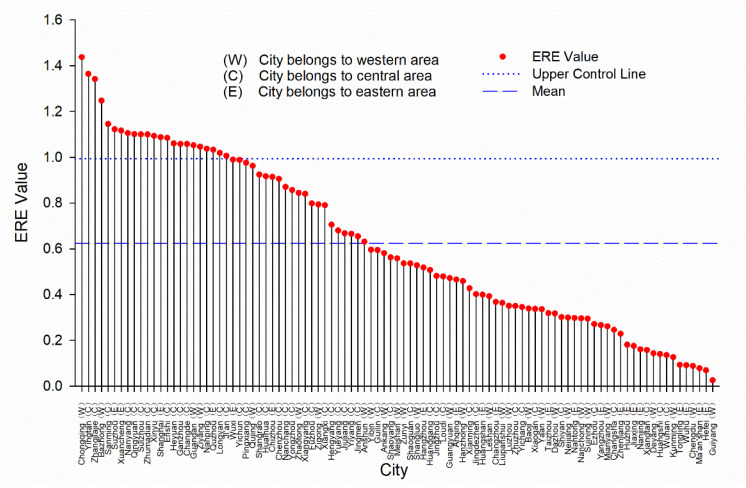
Histogram of super ERE of 97 cities in YRB during 2005–2016.

**Figure 4 ijerph-18-05697-f004:**
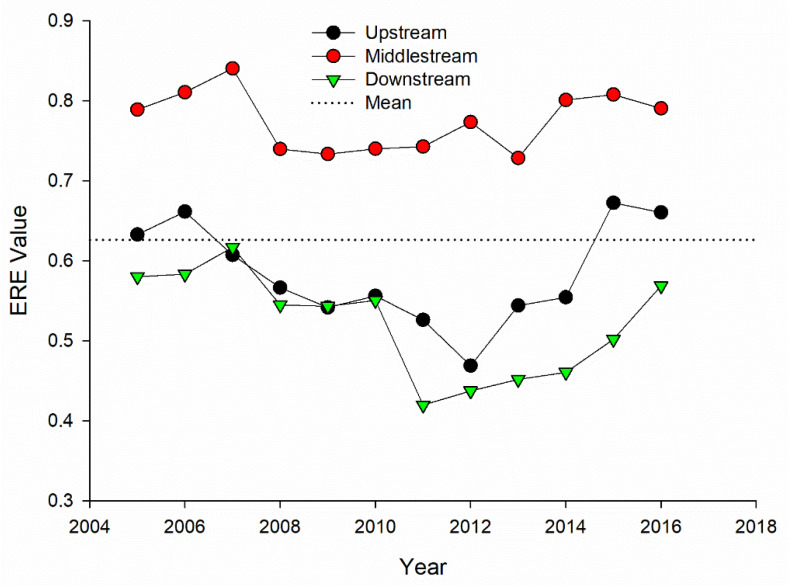
ERE in the upstream, middle stream, and downstream of YRB from 2005 to 2016.

**Figure 5 ijerph-18-05697-f005:**
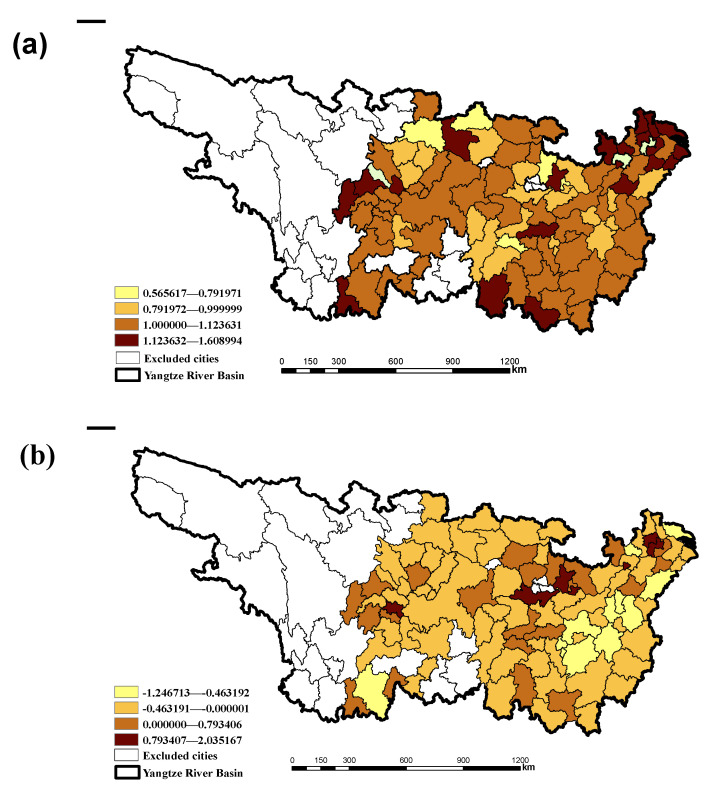
Total factor productivity (**a**) and its change (**b**) across 97 cities in the YRB during 2005–2016.

**Figure 6 ijerph-18-05697-f006:**
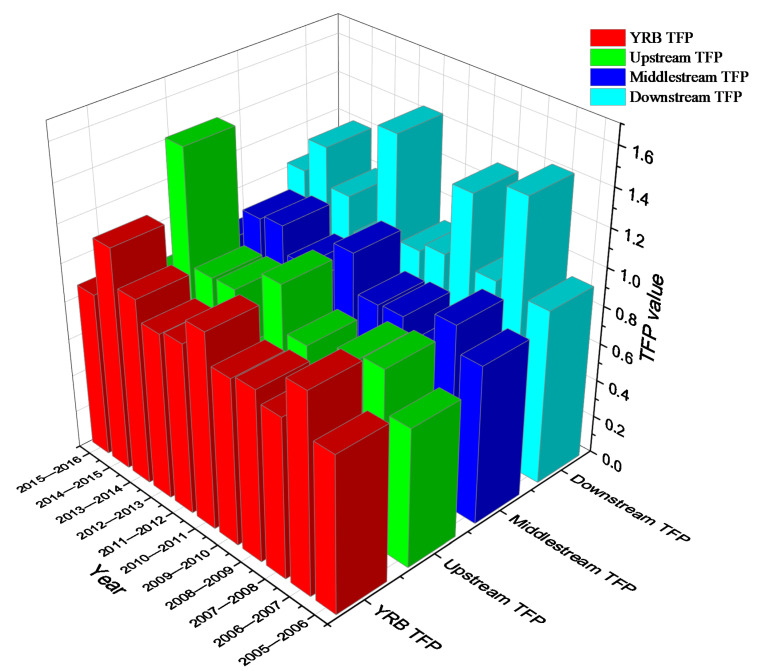
The trend of the Malmquist index of environmental regulation from 2005 to 2016.

**Figure 7 ijerph-18-05697-f007:**
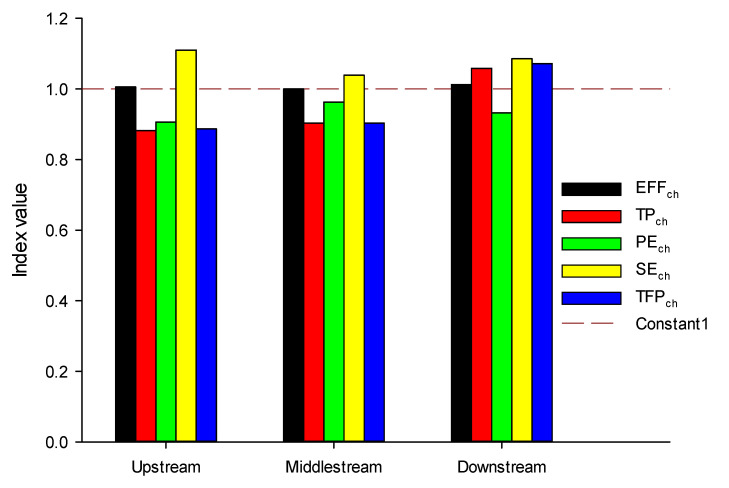
Total factor productivity decomposition in the upstream, middle stream, and downstream of Yangtze River Basin.

**Figure 8 ijerph-18-05697-f008:**
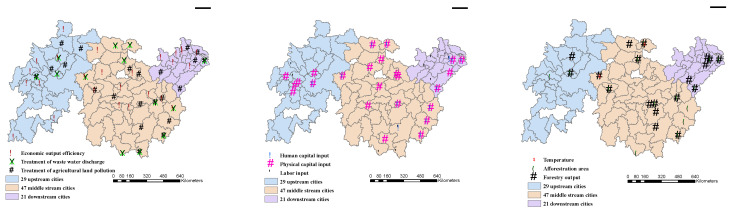
City distribution map with effective input and output.

**Table 1 ijerph-18-05697-t001:** Environmental regulation efficiency evaluation indices.

Vector	Serial Number	Index	Measurement	Unit	Data Source
Inputs	A1	Labor	Total employment	10, 000 persons	CCSY
	A2	Physical capital	Fixed asset investment	100 million yuan	CCSY
	A3	Human capital	Student enrollment by regular institutions of higher education	person	CCSY
Desirable outputs	B1	Economy Ecology	Real GDP	100 million yuan	CNKI, SY
	B2		Output of forestry	100 million yuan	CCSY
	B3		Artificial afforestation area	hectare	CFSY
Undesirable outputs	C1	Pollution	Industrial wastewater discharge	10,000 t/y	CRESY, CCSY
	C2		Consumption of chemical fertilizers (net)	10,000 t/y	CNKI, SY
	C3	Climate	Annual average temperature	°C	SY, CB

Explanatory notes: all measurements (e.g., GDP, employment, output) are at prefecture–city scale. CNKI: China National Knowledge Infrastructure; CCSY: China City Statistical Yearbook; CRESY: China Regional Economic Statistics Yearbook; CFSY: China Forestry Statistical Yearbook; SY: statistical yearbooks of each city and province; CB: climate bulletins of each city and province.

## Data Availability

The data presented in this study are available on request from the corresponding author. The data are not publicly available due to the unpublished articles based on these data.

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
