# Peer review of "Yangtze River Basin Environmental Regulation Efficiency Based on the Empirical Analysis of 97 Cities from 2005 to 2016"

_ijerph, 2021, doi:10.3390/ijerph18115697_

Round 1
Reviewer 1 Report
The publication of the manuscript is very interesting to central and local governmental engineers for designing the environmental urban planning.
Several revision requests are as follows
- The paper is an international journal-one. The readers are not familiar with the geology in China. Please make a map indicating the location of YRB inside Chinese territory.
- YRB is a basin? You mention the area surrounded by main stream and tributaries? Please indicate the definition of YRB.
- In the paper, YRB is divided into the western, central and eastern coastal areas. But the reader can not recognize the boundary. Maybe it is better for you to make a figure to express the boundaries of such three areas.
- Result Line 9; You use “RER”. This is wright?
- There are so many cities in tables. (for example; Figure 2) The reader has no image on the location of each city. At least, please mention the area (east, middle, west) which the city belongs.
- Page 9; Line 6: Why dose the 2008 Olympic Games make an influence to 2015-2016 declination of productivity?
- What purpose does the reference list between Appendix A and A-1 have?
Author Response
Please see the attachment.
Many thanks.

Reviewer 2 Report
The aim of the paper entitled Yangtze River Basin Environmental Regulation Efficiency Based on the Empirical Analysis of 97 Cities from 2005 to 2016 is to present the results of research of total factor productivity across 97 cities in the region of The Yangtze River Basin. Unfortunately, the main purpose of the article has not been clearly specified neither in the abstract nor in the introduction. In the introduction, there is no indication of the novelty of research, and the entire article looks more like a report than a research article. The article lacks a description of the research area, apart from the brief information contained in the introduction. The way of presenting the methodology used in the article is too general. The DEA-Malmquist Index also requires a detailed description. Please also correct the place where appendices are included.
Author Response
Please see the attachment.
Many thanks.

Round 2
Reviewer 1 Report
Still now, Figure 1 is not perfect. For example, Hunan territory should be colored in green. A slight modification and re-check for the figure-1 is necessary.
Reviewer 2 Report
Authors have corrected the manuscript and taken into account all remarks. Now, the paper can be accepted.